# Monitoring Gene Expression during a *Galleria mellonella* Bacterial Infection

**DOI:** 10.3390/microorganisms8111798

**Published:** 2020-11-16

**Authors:** Laura Moya-Andérico, Joana Admella, Rodrigo Fernandes, Eduard Torrents

**Affiliations:** 1Bacterial Infections and Antimicrobial Therapies Group, Institute for Bioengineering of Catalonia (IBEC), The Barcelona Institute of Science and Technology (BIST), 08028 Barcelona, Spain; jadmella@ibecbarcelona.eu (J.A.); rgaspar@ibecbarcelona.eu (R.F.); 2Microbiology Section, Department of Genetics, Microbiology, and Statistics, Faculty of Biology, University of Barcelona, 08028 Barcelona, Spain

**Keywords:** *Galleria mellonella*, *P. aeruginosa*, hemolymph, hemocytes, bioluminescence, promoter probe vector, optimized RNA extraction, ribonucleotide reductases

## Abstract

*Galleria mellonella* larvae are an alternative in vivo model that has been extensively used to study the virulence and pathogenicity of different bacteria due to its practicality and lack of ethical constraints. However, the larvae possess intrinsic autofluorescence that obstructs the use of fluorescent proteins to study bacterial infections, hence better methodologies are needed. Here, we report the construction of a promoter probe vector with bioluminescence expression as well as the optimization of a total bacterial RNA extraction protocol to enhance the monitoring of in vivo infections. By employing the vector to construct different gene promoter fusions, variable gene expression levels were efficiently measured in *G. mellonella* larvae at various time points during the course of infection and without much manipulation of the larvae. Additionally, our optimized RNA extraction protocol facilitates the study of transcriptional gene levels during an in vivo infection. The proposed methodologies will greatly benefit bacterial infection studies as they can contribute to a better understanding of the in vivo infection processes and pathogen–mammalian host interactions.

## 1. Introduction

*Galleria mellonella* (greater wax moth) is a popular animal model for the study of virulence and pathogenicity of different bacteria. The larvae are conveniently sized for manipulation, they do not require feeding, and they are inexpensive to purchase and breed. Moreover, they do not need any special infrastructure, they present a low biohazard risk, and they are more ethically accepted than the traditional murine models [1,2]. With an innate immune response very similar to the one found in mammals, *G. mellonella* larvae possess both cellular and humoral defenses [3]. Furthermore, they are also widely used for testing the efficacy and toxicity of new antimicrobial agents, thus offering an additional pre-screening model to lower the number of drug tests performed in mammalian animals [4,5]. Based on all these advantages, *G. mellonella* has recently become an attractive alternative infection model for gathering initial data.

Studying a bacterial infection in *G. mellonella* larvae can also be challenging due to some methodological limitations. The use of bacterial strains containing GFP (green fluorescent protein) vectors has been widely considered as GFP-tagged bacteria are easily visualized in confocal and fluorescent microscopes, thus allowing bacteria to be tracked within the hemolymph and hemocytes of the larvae [6,7]. The fluorescence of bacteria within infected larvae has also been measured with a spectrophotometer [8]. Unfortunately, most of the studies overlook that *G. mellonella* larvae possess natural green and red autofluorescence [8,9]. Therefore, the use of green and red fluorophores is associated with background autofluorescence problems that hinder the real measurement of bacterial fluorescence. For this reason, bioluminescence currently seems like the best methodology alternative to monitor an in vivo infection in *G. mellonella.* Bioluminescence is a rapid, cost-effective, and non-invasive imaging technique based on the synthesis of light derived from the *luxCDABE* operon’s presence in bacteria [10]. The use of vectors that can confer a genetically encoded luminescent phenotype to bacterial strains has already been reported in the literature [11,12].

*Galleria mellonella* has been recognized as a suitable model for studying *Pseudomonas aeruginosa* infections [13]. Currently one of the most multidrug-resistant pathogens alive, it is also a formidable opportunistic pathogen that can cause invasive and fulminant infections, such as acute pneumonia or bloodstream infections, especially in immunocompromised hosts. Remarkably, the same pathogen also causes chronic infections that can persist for months to decades, such as chronic lung infections in individuals who have a genetic disease such as cystic fibrosis [14]. *P. aeruginosa* divides rapidly during its infection, requiring a high and persistent supply of dNTPs to replicate its DNA quickly. Ribonucleotide reductases (RNRs) are key enzymes that catalyze the reduction of ribonucleotides to deoxyribonucleotides, thus providing the precursor molecules needed for DNA synthesis and repair. There are three RNR classes (I, II, and III), which are encoded by the *nrdAB*, *nrdJab*, and *nrdDG* genes, respectively. Curiously, *P. aeruginosa* is one of the few organisms that encode all three classes (Ia, II, and III) in its genome. Although how each RNR class works is not yet completely understood, it is established that a particularly tight regulation is required to guarantee bacterial viability. RNR gene expression is controlled by NrdR, a transcriptional factor that binds to NrdR boxes in the promoter regions of the different *nrd* genes and is key for the transcriptional regulation of RNRs [15,16].

Here, we report the construction and utilization of several *lux* vectors containing different *nrd* promoter fusions as a better approach to study and monitor a *P. aeruginosa* in vivo infection in *Galleria mellonella*. Additionally, we optimized a protocol for the isolation of total bacterial RNA during a *G. mellonella* infection. The optimized methodologies will be valuable for the study of different infections and further applications in *G. mellonella.*

## 2. Materials and Methods

### 2.1. Bacterial Strains, Plasmids, and Growth Conditions

The bacterial strains and plasmids used in this study are listed in Appendix A. *E. coli* and *P. aeruginosa* cells were routinely grown in Luria-Bertani (LB; Scharlab, Barcelona, Spain) medium at 37 °C. Antibiotics were added when required at the following concentrations: ampicillin, 50 µg/mL; and gentamicin, 10 µg/mL (*E. coli*) and 100 µg/mL (*P. aeruginosa*).

### 2.2. Recombinant DNA and PCR Methods

Recombinant DNA manipulations were performed using standard procedures [17] and following the manufacturers’ instructions. DNA fragments were amplified by PCR using Phusion™ High-Fidelity DNA Polymerase (Thermo Scientific, Madrid, Spain) for cloning and DreamTaq Green PCR Master Mix (Thermo Scientific, Madrid, Spain) for colony PCR. All primers used in this study are listed in Appendix A. Isolation of DNA fragments from agarose gels was carried out using the Monarch^®^ DNA Gel Extraction Kit (New England BioLabs, Ipswich, MA, USA). DNA fragments were digested with corresponding restriction enzymes (Thermo Scientific, Madrid, Spain) and ligated with T4 DNA ligase (Invitrogen, Carlsbad, CA, USA). Plasmid DNA was isolated using the GeneJET™ Plasmid Miniprep Kit (Thermo Scientific, Madrid, Spain) and quantified using a NanoDrop™ 1000 spectrophotometer (Thermo Scientific, Madrid, Spain). DNA was transformed into *P. aeruginosa* PAO1 cells by electroporation using a Gene Pulser Xcell™ electroporator (Bio-Rad, Hercules, CA, USA) as previously described [18].

### 2.3. Construction of Promoter Probe Vector with Bioluminescence Expression

Specific restriction site sequences were incorporated as needed at the 5′ ends of primers to facilitate cloning of fragments into the corresponding vector. The correct insertion of the fragments at each step was verified by colony PCR and DNA sequencing. The promoter probe vector pETS220-BIATlux (pETSlux) containing bioluminescence expression was made by first cloning the *Sac*I-*Kpn*I fragment containing the *luxCDABE* operon part of pMDM513 into the pBAM-Gm vector. From here, the *luxCDABE* genes (6109 bp, without promoter) were amplified by PCR using the Lux_SmaI_fw/pBAM-Gm up primers, and the amplicon was ligated into the pJET1.2 vector. Meanwhile, the pETS130 vector was digested with *Hind*III to remove the chloramphenicol acetyltransferase (*cat*) and GFP (mut3-GFP variant) genes. Both the resulting vector and the pJET1.2 vector containing the *luxCDABE* genes were digested with *Hind*III and ligated together to generate the pETSlux. The nucleotide sequence of the pETS220-BIATlux vector was deposited in GenBank (accession number MW117147).

### 2.4. Construction of nrd Promoter Fusions

Each *nrd* promoter region was amplified from *P. aeruginosa* PAO1 genomic DNA by PCR using a combination of forward and reverse primers that are listed in Appendix A. Each fragment (785 bp, 395 bp, 189 bp, 273 bp, and 333 bp for *nrdA*, *nrdJ*, *nrdD*, *nrdR*, and *anr*, respectively) was gel purified and cloned into the pJET1.2 vector. After digestion with *Eco*RI-*Sma*I for *nrdA*, *Sma*I-*Sac*I for *nrdJ*, and *Eco*RI-*Sac*I for *nrdD*, *nrdR*, and *anr*, fragments were ligated into pETSlux, generating the pETS221, pETS222, pETS223, pETS224, and pETS225 plasmids.

### 2.5. Galleria mellonella Maintenance and Injection

*G. mellonella* larvae were fed an artificial diet (15% corn flour, 15% wheat flour, 15% infant cereal, 11% powdered milk, 6% brewer’s yeast, 25% honey, and 13% glycerol) and reared at 34 °C in darkness as previously published [9]. To prepare the injection inoculum, the bacterial broth of the different strains was centrifuged at 4000× *g* rpm for 10 min. The supernatant was discarded, and the pellet was re-suspended in 5 mL of 1× PBS (Fisher Scientific, Madrid, Spain) during three washes. OD_590_ was measured using a 1/10 dilution with 1x PBS and equalized to a final OD_590_ of 1. Afterward, 10-fold serialized dilutions of the equalized culture were made with 1× PBS. *G. mellonella* larvae (200–220 mg each) were injected with 20–40 CFUs per larva through the top right proleg using a 26-gauge microsyringe (Hamilton, Reno, NV, USA). Groups of 3–6 larvae were injected for each condition and kept at 37 °C during the infection course.

### 2.6. Fluorescence Quantification and Microscopy

*G. mellonella* larvae were injected with an infective dose of *P. aeruginosa* PAO1 wild-type and *P. aeruginosa* PAO1 containing plasmids derived from pETS130 (Appendix A), which encode transcriptional fusions of the *nrdA* and *nrdJ* promoters with GFP and E2Crimson that were previously constructed in our laboratory (pETS134, pETS180, pETS226, and pETS227) [14,19]. As a control, a group of larvae was only injected with 1× PBS (Fisher Scientific, Madrid, Spain). The larvae were incubated at 37 °C for 16 h and until death (20 h for the infected larvae). After each time point, the larvae were anesthetized on ice for 10 min. Afterward, the tails of the larvae were cut off, and the hemolymph was collected into Eppendorf tubes while kept on ice to prevent melanization. The hemolymph from each group of larvae was pooled, and 1× PBS was added until 100 µL total volume was reached. Relative fluorescence was measured from the top of a 96-well black, flat-bottom microtiter plate (Fisher Scientific, Madrid, Spain) in an Infinite^®^ M200 Pro microplate reader (Tecan, Männedorf, Switzerland). The optimized Z-position was established as 18,000 μm, and the integration time was set as 20 µs. For GFP, the gain was adjusted to the optimal 142, the excitation wavelength was fixed at 480 nm, and the emission wavelength was set at 515 nm. For E2Crimson, the optimal gain was determined as 255 nm, and the excitation and emission wavelengths were established as 600 and 650 nm, respectively. The results obtained were analyzed using the GraphPad Prism 8.0 software (San Diego, CA, USA).

After fluorescence quantification, the hemolymph was visualized with an inverted fluorescence microscope ECLIPSE Ti−S/L100 (Nikon, Tokyo, Japan) coupled with a DS-Qi2 camera (Nikon, Tokyo, Japan) using a 100×/1.30 oil objective with the GFP and Texas Red filters for green and red fluorescence, respectively. The images obtained were analyzed using the NIS-Elements microscope imaging software (Nikon, Tokyo, Japan).

### 2.7. Bioluminescence Measurements

Relative luminescence of infected larvae was measured in a 6-well microtiter plate (Caplugs Evergreen, Buffalo, NY, USA) in a Spark^®^ multimode microplate reader (Tecan, Männedorf, Switzerland) with an integration time of 1000 ms. The results obtained were analyzed using GraphPad Prism 8.0 software (San Diego, CA, USA). The ImageQuant™ LAS 4000 mini imager (GE Healthcare, Chicago, IL, USA) allowed us to take chemiluminescence pictures of the worms over different time points of infection. Exposure time was optimized to 30 s, and images were afterward edited with ImageJ FIJI (Version 1.52p, National Institutes of Health, Bethesda, MD, USA). Before each measurement, larvae were anesthetized for ten minutes on ice.

### 2.8. RNA Extraction, Reverse Transcription, and Real-Time PCR

*G. mellonella* larvae were injected with an infective dose of *P. aeruginosa* PAO1 wild-type and incubated for about 16 h at 37 °C. Then, the larvae were anesthetized on ice for 10 min. Subsequently, the tails of the larvae were cut off using a size 23 sterile surgical blade (Paramount Surgimed, New Delhi, India), and the hemolymph was collected into Eppendorf tubes while kept on ice to avoid melanization. The hemolymph from each group of larvae was pooled, and hemocytes were pelleted by centrifuging at 1000× *g* for 5 min at 4 °C. The cell-free hemolymph was treated with RNAprotect^®^ Bacteria Reagent (Qiagen, Hilden, Germany), as stated in the manufacturer’s handbook for subsequent RNA purification. As a reference, *P. aeruginosa* PAO1 wild-type cells were grown in LB medium to mid-exponential growth phase (OD_550_ = 0.5) and then treated with RNAprotect^®^.

In both samples, the RNA was purified using the GeneJET™ RNA Purification Kit (Thermo Scientific, Madrid, Spain) according to the manufacturer’s instructions. DNase (Turbo DNA-*free*™, Applied Biosystems, Foster City, CA, USA) was used to remove DNA contamination that was verified by PCR. For cDNA synthesis, RNA was quantified using a NanoDrop™ 1000 spectrophotometer (Thermo Scientific, Madrid, Spain), and 0.25 µg of RNA was reverse transcribed using Maxima Reverse Transcriptase (Thermo Scientific, Madrid, Spain) along with random hexamer primers (Thermo Scientific, Madrid, Spain) by following the manufacturer’s guidelines.

Quantitative real-time PCR (qRT-PCR) was performed using PowerUp™ SYBR™ Green Master Mix (Applied Biosystems, Foster City, CA, USA) in a StepOnePlus™ Real-Time PCR System (Applied Biosystems, Foster City, CA, USA) according to the manufacturer’s protocol. All of the qRT-PCR reactions used specific gene primers that are listed in Appendix A. The *gapA* gene was used as an internal standard as its expression is vital during *P. aeruginosa* growth. Three replicates were performed for each sample, and the entire experiment was independently performed twice. The results were analyzed using the comparative Ct (cycle threshold) method (ΔΔCt) and plotted using GraphPad Prism 8.0 software (San Diego, CA, USA).

### 2.9. Other Methods

*G. mellonella* larvae were injected with an infective dose of *P. aeruginosa* PAO1 wild-type, *P. aeruginosa* PAO1 expressing GFP (PAO1::eGFP), and *P. aeruginosa* PAO1 expressing the *luxCDABE* genes (PAO1::lux) followed by incubation at 37 °C for about 19 h or until death. The larvae infected with PAO1 wild-type and PAO1::eGFP were imaged in a Gel Doc™ XR+ imaging system (Bio-Rad, Hercules, CA, USA) using the Alexa Fluor™ 488 protocol with UV trans-illumination. The same larvae were also imaged directly under an LSM 800 confocal laser scanning microscope (Zeiss, Jena, Germany) with the 20×/0.8 air and 63×/1.4 oil objectives. The larvae infected with PAO1 wild-type and PAO1::lux were visualized in the ImageQuant™ LAS 4000 mini (GE Healthcare, Chicago, IL, USA) and Odyssey^®^ Fc (LI-COR, Lincoln, NE, USA) imaging systems using the chemiluminescence method with a 30-s exposition time. In the LAS 4000 mini, the larvae were also pictured using the digitization (epi-illumination) method with an exposure time of 1/30 s. As a reference, free bacteria grown in LB media were also imaged with the same methodologies. Images were analyzed using ImageJ FIJI (Version 1.52p, National Institutes of Health, Bethesda, MD, USA) and GraphicConverter (Version 11, Lemke Software, Peine, Germany).

## 3. Results and Discussion

### 3.1. Optimization of Reporter Genes for Gene Expression Studies

To first determine the best reporter gene for bacterial gene expression studies in *G. mellonella*, larvae were infected with two *P. aeruginosa* PAO1 strains, each harboring either green fluorescent protein (GFP) or luciferase genes (*lux*) inserted in its chromosome (PAO1::eGFP and PAO1::lux, respectively). These were chosen as they represent the most commonly used non-enzymatic reporter genes. The larvae were incubated at 37 °C until their death, which is when sufficient bacterial load should be available to detect its fluorescence or bioluminescence. Then, the larvae were visualized in imaging systems readily available in the laboratory.

The first instrument tested was the Gel Doc™ imaging system (see Materials and Methods). This instrument or an analog is readily available in most laboratories as it is commonly used to image protein and nucleic acid gels, blots, and macroarrays. This system also supports multiple detection methods, including fluorescence and colorimetric detection [20]. Using the preset Alexa Fluor™ 488 protocol, *P. aeruginosa* PAO1 WT and PAO1::eGFP on LB plates were first imaged. As seen in Figure 1A, green luminescence was seen with PAO1::eGFP but not with the wild-type strain as it is not expressing a fluorescent protein. The next step was to image larvae infected with the two previous strains to see whether the same result could be obtained. Unfortunately, both groups of larvae showed some green fluorescence around the body’s edges that is most likely autofluorescence. Specific bacterial fluorescence was not seen inside the larvae as the fluorescence probably cannot be detected through the larval cuticle. Additionally, we tried to image the same samples under a confocal microscope. Entire larvae were visualized using the green (488 nm) laser for PAO1::eGFP and the differential interference contrast (DIC) mode for non-fluorescent bacteria (PAO1 WT). As seen in Figure 1B, no bacterial fluorescence could be seen as the laser could not penetrate the cuticle of the larvae. Instead, only green autofluorescence was again detected in both types of samples. In all cases, the use of protein fluorescence was not the best option for monitoring gene expression during a *G. mellonella* infection as it cannot overcome the larval cuticle. Furthermore, as seen in our experiments, the particularly high autofluorescent background of the worms can hinder the precise quantification of the protein.

As the fluorescence detection systems were not successful, the next step was to try using bioluminescence. This methodology is commonly used to study ongoing biological processes in living organisms [21]. *P. aeruginosa* PAO1 WT and PAO1::lux on LB plates were first imaged in two different devices, the ImageQuant™ LAS 4000 mini and the Odyssey^®^ Fc imaging systems (see Materials and Methods). These two instruments or equivalent ones are commonly found in laboratories as they are used to analyze chemiluminescent Western blots, among other applications. In both systems, the LB plate with *P. aeruginosa* PAO1::lux showed a high amount of bioluminescence while the PAO1 WT plate did not. The same result was obtained with the larvae infected with *P. aeruginosa* PAO1::lux and PAO1 WT (Figure 1C). One finding worth emphasizing is that the larvae that were not infected with bacteria expressing the *lux* genes had almost negligible background bioluminescence. Thus far, it appears that bioluminescence is the best method for studying bacterial infections using whole larvae.

### 3.2. Determining Autofluorescence in Galleria mellonella

As was clearly shown previously, the larval cuticle is impermeable to the light emitted by the different fluorescent proteins used in this work. As a way to overcome the cuticle’s opacity, further experiments were done to try to measure the signal emitted directly from fluorescent proteins present in the hemolymph extracted from infected larvae. For this, *G. mellonella* larvae were injected with *P. aeruginosa* PAO1 wild-type (WT), two *P. aeruginosa* PAO1 strains expressing GFP (P*nrdA*-GFP [pETS134] and P*nrdJ*-GFP [pETS180]), two *P. aeruginosa* PAO1 strains expressing E2Crimson (P*nrdA*-E2Crimson [pETS226] and P*nrdJ-*E2Crimson [pETS227]), and PBS. The hemolymph from each group was extracted, and then, green and red fluorescence was measured in a Tecan plate reader using GFP (Figure 2A) and E2Crimson (Figure 2B) settings, respectively. At 16 and 20 h post-infection, *P. aeruginosa* PAO1 WT presented an inherent amount of both green and red autofluorescence when compared to the bacterial strains expressing GFP and E2Crimson (Figure 2A,B). Similar results were also seen with bacteria expressing mOrange and eYFP (data not shown). At 16 and 20 h post-injection, the hemolymph of larvae injected with PBS also had high autofluorescence with both GFP and E2Crimson that resulted in values comparable to the ones obtained with the expression vectors. The only difference between non-infected and infected larvae is that PAO1 WT causes hemocyte proliferation within the larvae in response to the infection [22], so it seems that hemocytes also possess intrinsic autofluorescence. Additionally, it can be noticed that the P*nrdA* strains show more fluorescence than the P*nrdJ* strains, but this is related to their genetic expression during infection, as we previously reported [14]. When quantifying GFP and E2Crimson fluorescence, all strains follow the same pattern, and it is undeniable that *G. mellonella*’s autofluorescence cannot pass unnoticed any longer.

Additionally, the same hemolymph was visualized under a fluorescence microscope to reinforce the plate reader results. In Figure 2C, the hemolymph of larvae infected with *P. aeruginosa* PAO1 WT was used to observe the natural green and red autofluorescence of hemocytes. Hemolymph containing *P. aeruginosa* PAO1 P*nrdA*-GFP appears in Figure 2D, where bacteria can be seen expressing green fluorescence along with the green autofluorescence of hemocytes. The same happens in the hemolymph containing *P. aeruginosa* PAO1 P*nrdA*-E2Crimson, as seen in Figure 2E, where bacteria and hemocytes appear red when merging the phase contrast and red channels.

Fluorescent promoter probe vectors are widely used for gene expression analysis under different conditions [23,24], so these vectors would be ideal to use in a *G. mellonella* infection. However, the results obtained with the hemolymph studies confirmed that fluorescent protein vectors could not be used in *G. mellonella*. The autofluorescence of hemocytes during *P. aeruginosa* infections poses a critical limitation in using fluorescent vectors as changes in fluorescence expression cannot be easily distinguished. For this reason, vectors expressing bioluminescence seem like the most optimal alternative.

### 3.3. Construction of pETS220-BIATlux Vector and nrd Fusions

To study gene expression using bioluminescence in *G. mellonella* larvae, a vector containing the *lux* genes was first needed. The approach was to create a promoter probe vector that could be used with any gene promoter and not just a vector containing the promoters required for our purpose. For this, the pETS130 vector [14] was modified by replacing the *cat* and mut3-GFP variant genes with the *luxCDABE* genes, thus generating the pETS220-BIATlux (pETSlux) vector with a final size of 10,677 base pairs (see Materials and Methods). This plasmid (Figure 3) offers several advantages: (1) it has a multi-cloning site (MCS) containing unique restriction sites for easy insertion of the promoter fragment of interest, (2) it carries gentamicin resistance for simple selection, and (3) it is a broad-host-range vector as it contains both pBBR1 *oriV* and pBBR1 *Rep* that allow its replication in a variety of microorganisms [25].

Once the vector was successfully constructed, the promoters of interest were cloned using the unique cutter restriction enzymes within the multi-cloning site. As a proof of concept, the promoters of the different *P. aeruginosa* ribonucleotide reductase genes (*nrd*) (P*nrdA*, P*nrdJ*, P*nrdD*, and P*nrdR*) were all included to efficiently detect differences in gene expression during infection. These genes were previously characterized by our group during infection in *Drosophila melanogaster* [14] and *Danio rerio* [24]. As a negative control, a fragment of the *anr* gene similar in size to the promoters used was inserted in the vector using the MCS. This was done to detect the vector’s intrinsic levels of bioluminescence. After all the *nrd* promoter fusions were successfully constructed, the vectors were electrotransformed into *P. aeruginosa* PAO1 wild-type cells. Although the vector is relatively large (>10 kb), all the DNA manipulations were carried out effortlessly, and the vector was easily electroporated into the PAO1 cells in the first attempt. Finally, positive clones were selected using gentamicin. Using the pETSlux for cloning the different promoters facilitated the process in many ways: the promoters were easily cloned using the MCS, the promoter fusions could be cultured in both *E. coli* and *P. aeruginosa*, and positive clones were readily detected using gentamicin selection. These advantages demonstrate the high practicality of the pETSlux vector in bacterial gene expression studies.

### 3.4. Monitoring a P. Aeruginosa Infection in G. Mellonella Larvae

As proof of concept for our bioluminescence vector, we evaluated the expression of the well-known ribonucleotide reductase (*nrd*) genes [15,16] to track an in vivo infection. *G. mellonella* larvae were injected with the *P. aeruginosa* PAO1 wild-type strains containing the different *nrd* promoter-*lux* fusion constructions. Relative luminescence was measured for each larval group at several time points during the infection course: 8, 14, 17, and 20 h post-infection with the latter corresponding to larval death. The bioluminescence within the larvae should vary depending on the level of expression of the different *nrd* genes as regulated by the promoters controlling the *lux* operon. In effect, the infection could be monitored at different time points as the relative luminescence units (RLU) within the larvae increased over time for the different strains (Figure 4A). The expression of both P*nrdR*-lux and P*nrdJ*-lux started rising at 14 h post-infection (4059 and 2740 RLU, respectively) and continued escalating throughout the remainder of the infection. In contrast, P*nrdD*-lux and P*nrdA*-lux both had a lower expression that did not reach similar levels as P*nrdJ*-lux (about 2.5 × 10^5^ RLU) until the larvae were dead. At 20 h post-infection (death), P*nrdR*-lux had the highest RLU values (3.5 × 10^6^), thus indicating that *nrdR* is highly expressed not only at death but throughout the whole infection. The negative control, *Anr*-lux, had low levels of bioluminescence that ranged from 36 to 2600 RLU, which were consistent with background signals caused by leaking reporter expression. Despite the leaking, the values of all the other strains were much higher than *Anr*-lux at all time points. The highest RLU value obtained with *Anr*-lux was only 2600 at 20 h post-infection, which is about 1340 times less than the corresponding value for P*nrdR*-lux (3.5 × 10^6^ RLU).

Bioluminescence images were also taken at 17 and 20 h post-infection for each of the larvae conditions using the ImageQuant™ LAS 4000 mini imaging system (Figure 4B). The images show that the larvae emitted more or less bioluminescence according to their respective *nrd* expression (depending on the RNR class expressed), which coincided with the results obtained in Figure 4A. Additionally, the induction in the expression of the different *nrd* genes during the infection was quantified (Figure 4C). *Anr*-lux (negative control) displayed a luminescent background signal, as previously mentioned, which was subtracted from each of the strains and time points. Therefore, the values that appear for each strain in Figure 4C correspond to its induction factor of expression compared to itself at the initial stage of infection (8 h post-infection). A high induction in *nrdR* expression was distinctly noticed during all time points (119, 18,217, and 106,174-fold induction at 14, 17, and 20 h, respectively), followed by *nrdJ* (73, 3988, and 6706-fold induction) and *nrdD* (12, 543, and 3557-fold induction) at the same time points. On the other hand, a small induction (4–46-fold) in *nrd*A expression was seen in the first hours until it suddenly increased at the last time point (up to 6838 times). However, this induction value was still considerably smaller than the one seen with the *nrdR* gene (106,174) at the same time point. A broadened view of Figure 4 confirmed the shift of *nrd* expression during a *P. aeruginosa* infection in *G. mellonella* larvae that was possible to observe and monitor due to the *lux* constructions. Furthermore, the shift seen corresponds with previous results obtained in our lab. In the *Drosophila melanogaster* and *Danio rerio* infection models, the expression of *nrdJ* and *nrdD* (RNR class II and III, respectively) was also highly induced during the course of a *P. aeruginosa* infection [14,24]. This demonstrates the importance of anaerobic environments for the in vivo expression of these RNR classes [14,16,26].

Another approach for measuring gene expression during infection in *G. mellonella* was attempted using qRT-PCR. For this, a protocol to extract RNA from the bacterial cells inside the *G. mellonella* larvae was initially optimized (Figure 5A). First, larvae were injected with an infection dose of *P. aeruginosa* PAO1 wild-type (WT) and then kept at 37 °C for about 16 h. This time point was chosen based on previous growth curves done with PAO1 WT in *G. mellonella* that revealed that at 16 h post-infection, close to 10^8^ bacterial cells per milliliter were present within the larvae (Figure 5B). According to the GeneJET™ RNA purification kit protocol, a bacterial concentration near 10^9^ is recommended for optimal RNA yield. After incubation, the larvae were anesthetized on ice for 10 min before cutting their tail off to extract the hemolymph. The hemolymph from each larvae group was pooled and then centrifuged at low speed for 5 min at 4 °C to remove the hemocytes. Up to this point, it is important to maintain the hemolymph on ice to prevent melanization. The cell-free hemolymph containing PAO1 WT was treated with RNAprotect^®^ and afterward used for RNA extraction, cDNA preparation, and qRT-PCR analysis. By using this optimized protocol, *P. aeruginosa* PAO1 cells that were infecting *G. mellonella* larvae in vivo were able to be isolated for efficient RNA purification, which yielded high and pure RNA concentrations (Appendix A) that were successfully used in downstream applications.

The RNA expression of the infection cells was analyzed relative to the gene expression in *P. aeruginosa* PAO1 wild-type cells grown in LB medium to a mid-exponential growth phase. This control was selected based on the growth curve seen in Figure 5B that shows that the bacterial cells within the larvae are in the mid-exponential growth phase during sample collection. The growth curve also indicates that an active DNA synthesis is underway due to the involvement of RNR during the infection process. To ensure the reaction specificity of the qRT-PCR results, a melting curve analysis was performed, which showed a single, pure amplicon for each of the genes assessed (Appendix A). The results were then analyzed using the comparative Ct (cycle threshold) method (∆∆Ct) by comparing the Cts of the different genes in a PAO1 WT infection against the Cts of the same genes in a PAO1 WT exponential culture, and the transcript levels of each gene were normalized using the *gapA* internal control. Differences in the expression of the different *nrd* genes could be clearly seen (Figure 5C). The highest inductions were seen with *nrdR* and *nrdJ,* which were 7.7 and 7.5 times, respectively, more induced in infection than in planktonic culture. Less induction was seen with *nrdA* (3.1-fold change), while *nrdD* only had a 1.7-fold change induction. A 5.3-fold increase in the expression of the *norC* gene, a marker of anaerobiosis, indicated a decrease in oxygen levels, leading to a shift in the anaerobic metabolism of PAO1 WT during infection. When compared to the results obtained with the different pETSlux constructions, the RNR expression pattern is very similar except for *nrdD* and *nrdA* since P*nrdD*-lux had higher expression than P*nrdA*-lux. It is important to keep in mind that the RNA was extracted about 1 h before the time point used in the pETSlux measurements, and gene expression depends on the metabolic state of the bacteria at the time of RNA extraction. Therefore, measurements using pETSlux are more efficient since gene expression can be measured in vivo at multiple time points (independent of bacterial concentration inside the larvae) without much manipulation of the larvae or bacteria. Nevertheless, both the bioluminescence and qRT-PCR experiments revealed that *nrdR* is significantly induced during a *P. aeruginosa* infection. Although *nrdR* expression was reported to increase in vitro when interacting with T84 cells [26], this is the first time that *nrdR* induction is reported in an in vivo infection experiment. Nevertheless, further work is needed to elucidate its biological role under these specific conditions.

In this study, we demonstrated the limitation with the use of fluorescent vectors in *Galleria mellonella* due to its intrinsic autofluorescence. For this reason, a promoter probe vector containing bioluminescence expression was constructed that offers a simple and effective method for monitoring gene expression in vivo using the *G. mellonella* animal model of infection. Furthermore, an RNA extraction protocol for bacterial cells from within *G. mellonella* larvae was optimized to enable the study of transcriptional levels of genes during an in vivo infection. These findings will allow for a better understanding of the in vivo infection process of different bacteria in a setting that mimics mammalian body temperature and pathogen–mammalian host interactions that can only be achieved in an economical and ethical manner when using the *G. mellonella* model.

## Figures and Tables

**Figure 1 microorganisms-08-01798-f001:**
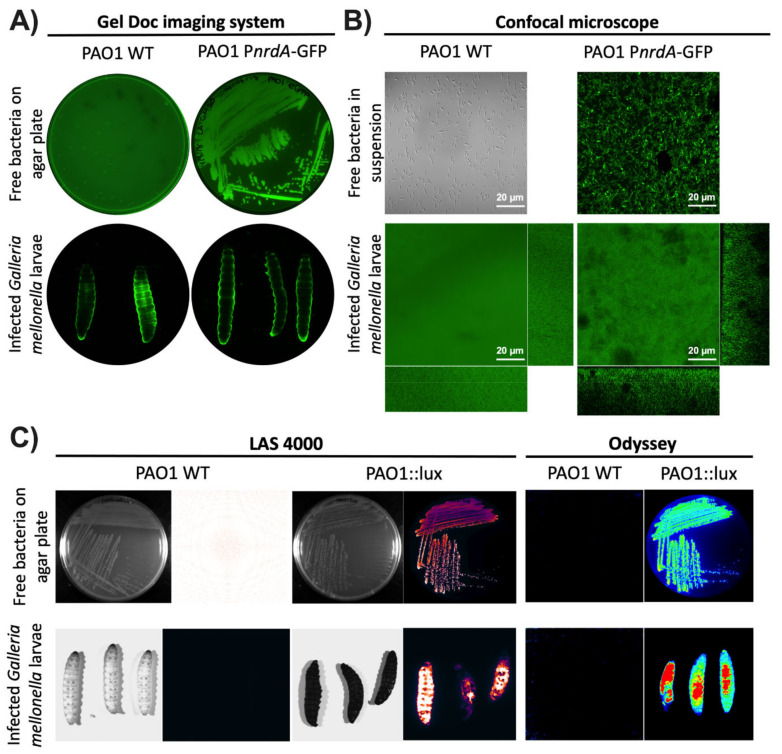
Visualization of *G. mellonella* larvae using different imaging methodologies. *P. aeruginosa* PAO1 wild-type (WT) and PAO1::eGFP strains were imaged using (**A**) the Gel Doc™ imaging system and (**B**) the LSM 800 confocal microscope. Free bacteria were imaged on LB media as well as inside *G. mellonella* larvae. (**C**) *P. aeruginosa* PAO1 wild-type (WT) and PAO1::lux were imaged as free bacteria on an LB plate and within *G. mellonella* larvae using the ImageQuant™ LAS 4000 mini and Odyssey^®^ Fc imaging systems.

**Figure 2 microorganisms-08-01798-f002:**
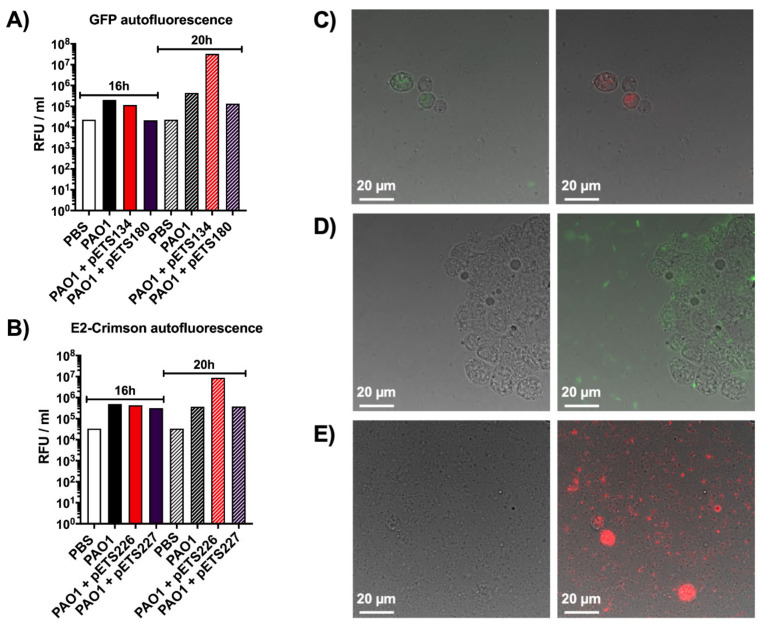
Autofluorescence studies in the hemolymph of *G. mellonella*. (**A**) Quantification of green autofluorescence, in relative fluorescence units per milliliter of hemolymph, using larvae infected with different *P. aeruginosa* strains and PBS (negative control) at 16 and 20 h post-infection. (**B**) Quantification of red autofluorescence, in relative fluorescence units per milliliter of hemolymph, using larvae infected with different *P. aeruginosa* strains and PBS (negative control) at 16 and 20 h post-infection. (**C**) Phase contrast merged with green and red fluorescence images of hemolymph extracted from larvae infected with *P. aeruginosa* PAO1 wild-type showing the green and red autofluorescence of hemocytes, respectively. (**D**) Phase contrast only and phase contrast merged with green fluorescence images of hemolymph extracted from larvae infected with *P. aeruginosa* PAO1 P*nrdA*-GFP. (**E**) Phase contrast only and phase contrast merged with red fluorescence images of hemolymph extracted from larvae infected with *P. aeruginosa* PAO1 P*nrdA*-E2Crimson.

**Figure 3 microorganisms-08-01798-f003:**
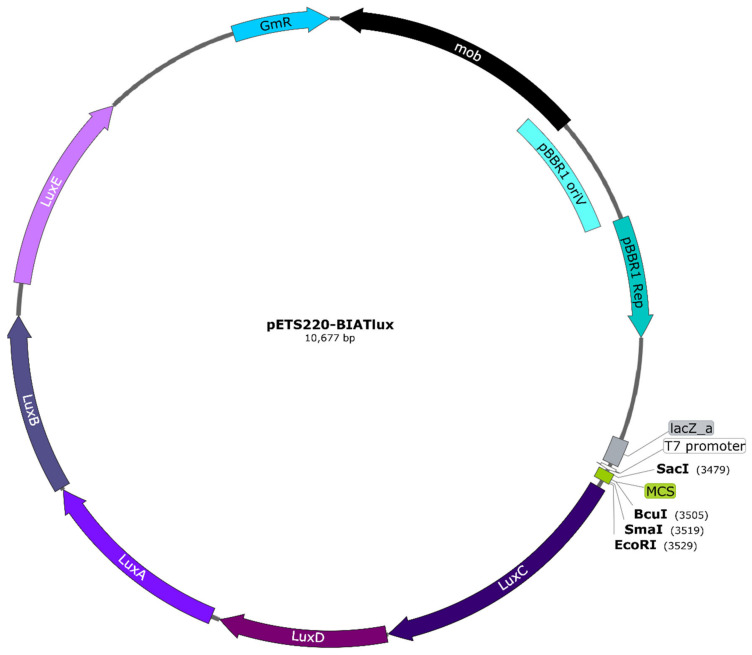
Map of the pETS220-BIATlux (pETSlux) promoter-probe vector. The *luxCDABE* genes were inserted into the backbone of the pETS130 vector. Relevant genetic elements include: gentamicin resistance (Gm^R^) imparted by *aacC1*, plasmid mobilization functions encoded by *mob,* and pBBR1 *oriV* and pBBR1 *Rep* as the replication origin and replication protein, respectively, that are essential for broad-host-range capability. Unique cutter restriction enzymes within the multi-cloning site (MCS) are shown in bold. All genes are represented in scale according to the total length of the plasmid. The vector map was designed with SnapGene^®^ version 5.0.8 (GSL Biotech, San Diego, CA, USA).

**Figure 4 microorganisms-08-01798-f004:**
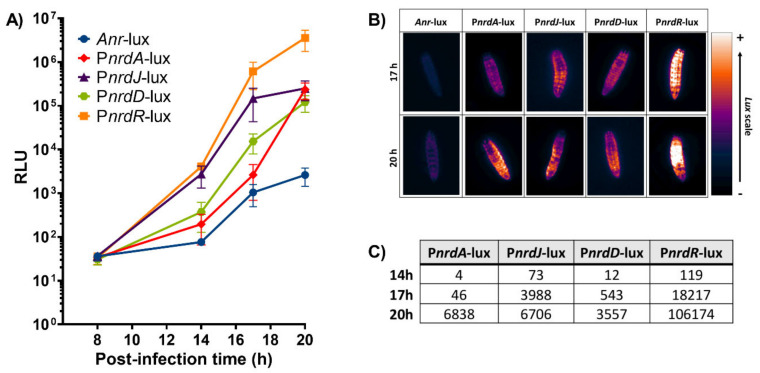
Study of a *P. aeruginosa* PAO1 infection in *G. mellonella* larvae using the different *lux* constructions. (**A**) Measurements of relative luminescence (RLU) in *G. mellonella* during several time points after infection within the different strains. (**B**) Images of *G. mellonella* larval bioluminescence taken with the ImageQuant™ LAS 4000 mini imager at 17 and 20 h post-infection and visualized using the Gem lookup table from ImageJ FIJI. (**C**) Bioluminescence induction factors during the different time points (14, 17, and 20 h post-infection). *Anr*-lux background was subtracted from each of the strains.

**Figure 5 microorganisms-08-01798-f005:**
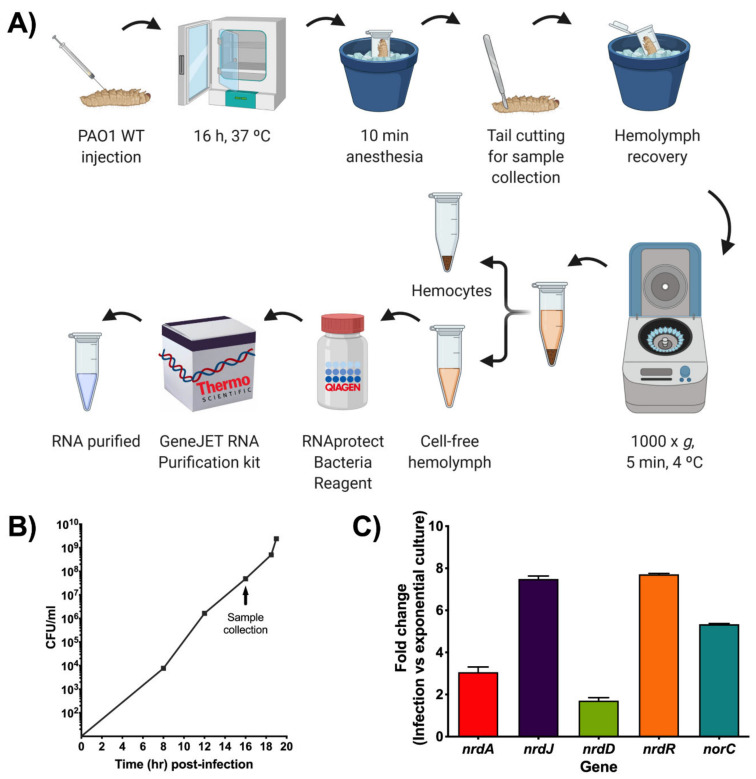
Gene expression studies during a *P. aeruginosa* infection in *G. mellonella*. (**A**) Schematic representation of the steps involved in the optimized RNA extraction protocol for bacterial cells derived from a *G. mellonella* infection. Created with BioRender.com. (**B**) *P. aeruginosa* PAO1 wild-type (WT) growth curve in *G. mellonella* larvae that was determined by calculating CFU/mL at different time points after infection as well as at death (last point). Samples for qRT-PCR were taken when the bacteria were in the exponential growth phase (close to 10^8^ CFU/mL) as indicated by the arrow. (**C**) Fold change of *nrd* and anaerobic (*norC*) genes determined by qRT-PCR during PAO1 WT infection compared to PAO1 WT cells grown to exponential phase in LB medium. The *gap* gene was used as an internal standard. The values shown are the average from two independent experiments, and the error bars indicate a positive standard deviation.

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
