# Peer review of "Monitoring Gene Expression during a Galleria mellonella Bacterial Infection"

_microorganisms, 2020, doi:10.3390/microorganisms8111798_

Round 1
Reviewer 1 Report
Minor revision
Basically, I think it is quite a remarkable piece of work that you did and I suggest this manuscript could get published. Galleria mellonella larvae has become a popular model for studying virulence and pathogenicity of different bacteria. In the current manuscript authors constructed and investigated a set of new vectors with bioluminescence expression that can replace traditional fluorescent protein vectors to omit the problem of larval autofluorescence. They also optimized RNA extraction protocol to further promote gene expression studies in this model system. Overall, the MS is well written, the experiments were well performed, the data is of high quality and well presented. I would recommend it for publication after the authors address the following minor issues.
Introduction
Lines 34-37. “With an innate immune response very similar to the one found in mammals, G. mellonella larvae possess both cellular and humoral defenses. Furthermore, they are also widely used for testing the efficacy and toxicity of new antimicrobial agents, thus offering an additional pre-screening model to lower the number of drug tests performed in mammalian animals.” You should add relevant references to support these two statements.
Materials and Methods
Line 127. According to provided references, authors utilized mutant of GFP with enhanced fluorescence (mut3-gfp variant) as a reporter system. Please, check and provide this information.
Results and Discussion
Line 255. At what time point autofluorescence of larval hemolymph injected with PBS was measured? Please indicate this information in text and in Fig.2.
Fig.5C. Please, add statistical evaluation of data.
Author Response
November 11, 2020
Dear Reviewer,
Our responses to your comments are detailed below.
With the manuscript changes detailed below and our answers to the reviewers’ comments, I hope that you will now find the revised version of our manuscript acceptable for publication in Microorganisms.
Sincerely,
Dr. Eduard Torrents
Reviewer #1's comments:
Introduction
Lines 34-37. “With an innate immune response very similar to the one found in mammals, G. mellonella larvae possess both cellular and humoral defenses. Furthermore, they are also widely used for testing the efficacy and toxicity of new antimicrobial agents, thus offering an additional pre-screening model to lower the number of drug tests performed in mammalian animals.” You should add relevant references to support these two statements.
Thank you for this comment. We have added the references after these two statements (line 36 and 39).
Materials and Methods
Line 127. According to provided references, authors utilized mutant of GFP with enhanced fluorescence (mut3-gfp variant) as a reporter system. Please, check and provide this information.
Thank you for pointing this out. The reviewer is correct, a mut3-GFP variant was used. This information was added to the manuscript (line 107 and 303) and in Table S1 (description for pETS130-GFP).
Results and Discussion
Line 255. At what time point autofluorescence of larval hemolymph injected with PBS was measured? Please indicate this information in text and in Fig.2.
The autofluorescence of larval hemolymph injected with PBS was measured at 16h and 20h post-injection. Since both measurements yielded the same values, we only added one bar in the graph. But the reviewer is correct, it is not clear in the graph or in the text. For this reason, we added a second bar under 20h in Figures 2A and 2B and updated the image in the revised manuscript. Also, a sentence clarifying the time points was added in lines 261-262 and in the caption for Figure 2A (line 274) and Figure 2B (line 277).
Fig.5C. Please, add statistical evaluation of data.
Thank you for pointing this out. Comments referring to statistical analysis have been added to the text (line 183-184) and in the caption of figure 5C (lines 438-439).

Reviewer 2 Report
The study was focused on the construction of a promoter probe vector with bioluminescence expression as well as the optimization of a total bacterial RNA extraction protocol to enhance the monitoring of in vivo infections. By employing the vector to construct different gene promoter fusions, variable gene expression levels were efficiently measured in G. mellonella larvae at various time points during the course of infection and without much manipulation of the larvae. The proposed methodologies will benefit bacterial infection studies as they can contribute to a better understanding of the in vivo infection processes and pathogen-mammalian host interactions.
In my opinion, the manuscript is valuable, however, some improvements are recommended:
- Authors should place electropherograms of RNA samples evidencing the high purity and integrity should be added in the Results section or Supplementary file - integrity and purity of RNA samples are crucial RT-qPCR studies,
- Quantitative RT-PCR: Were primers optimized to yield 95%-100% of PCR reaction efficiency?
- Authors used SYBR Green fluorescent dye during gene expression studies. In this case, it is obligatory to perform Melting Curve Analysis, and results of this examination should be added in the manuscript or Supplementary file (e.g., JPG or TIFF file),
- Discussion and interpretation of the results are surprisingly superficial and should be thoroughly revised,
-I noticed few style and grammar errors, hence, I recommend profound revision of the English language usage by a native speaker.
Author Response
November 11, 2020
Dear Reviewer,
Our responses to your comments are detailed below.
With the manuscript changes detailed below and our answers to the reviewers’ comments, I hope that you will now find the revised version of our manuscript acceptable for publication in Microorganisms.
Sincerely,
Dr. Eduard Torrents
Reviewer #2's comments:
- Authors should place electropherograms of RNA samples evidencing the high purity and integrity should be added in the Results section or Supplementary file - integrity and purity of RNA samples are crucial RT-qPCR studies.
Thank you for the suggestion. We have added a table in the supplementary file (Table S3) with the Nanodrop values as evidence of the good quality of the RNA samples. The table has also been referenced in the results section (lines 399-400).
- Quantitative RT-PCR: Were primers optimized to yield 95%-100% of PCR reaction efficiency?
The PCR reaction efficiencies of the qRT-PCR primers used in this study were not formally determined. However, primers were designed with the utmost care to ensure maximum specificity and efficiency. Primers were designed using PrimerBLAST in Thermodynamic Alignment Mode. Primer specificity was checked with PrimerBLAST and MFE Primer1 (using the whole Pseudomonas genus as a reference). Primer specificity was verified by the absence of secondary bands in agarose gel electrophoresis and secondary peaks in the melting curves (melting curves showing single peaks can be seen in Figure S1).
Concerning the thermodynamical properties of the primers, NetPrimer (PREMIER Biosoft) and MFE Primer were used. In compliance with the “Universal Thermal Cycling Conditions” (Applied Biosystems), only primers with a predicted melting temperature of 60-65 ºC were accepted (average of the Tms provided by different tools). The ΔΔG of target binding, self-dimerization and cross-dimerization were checked, and only primers for which dimerization ΔΔGs were smaller than a third of that of the intended target binding were accepted. Additionally, we looked for potential hairpins using the OligoAnalyzer tool (Integrated DNA Technologies). We rejected primers if the melting temperatures of any of the predicted hairpins were not at least 10 ºC lower than the intended targets’ Tm.
1 Wang K, Li H, Xu Y, Shao Q, Yi J, Wang R, et al. MFEprimer-3.0: quality control for PCR primers. Nucleic Acids Research. 2019;47(W1):W610-W3.
- Authors used SYBR Green fluorescent dye during gene expression studies. In this case, it is obligatory to perform Melting Curve Analysis, and results of this examination should be added in the manuscript or Supplementary file (e.g., JPG or TIFF file).
The reviewer is correct, the melting curve analysis has been added as supplementary figure 1 (Figure S1). Also, a few lines have been added to refer to this analysis (lines 406-408).
- Discussion and interpretation of the results are surprisingly superficial and should be thoroughly revised.
Thank you for pointing this out. Several sentences have been added to extend the discussion and interpretation further (lines 292-293, 324-325, 358, 373-376, 404-406, 427-428, and 440-441).
-I noticed few style and grammar errors, hence, I recommend profound revision of the English language usage by a native speaker.
Thank you for this recommendation. A native speaker has proofread the paper and newly corrected sentences and words are highlighted with the “Track Changes” function throughout the manuscript.
